# Cationic PLGA Nanoparticle Formulations as Biocompatible Immunoadjuvant for Serum Production and Immune Response against *Bothrops jararaca* Venom

**DOI:** 10.3390/toxins14120888

**Published:** 2022-12-19

**Authors:** Emanuell dos Santos-Silva, Manoela Torres-Rêgo, Fiamma Gláucia-Silva, Renata Carvalho Feitosa, Ariane Ferreira Lacerda, Hugo Alexandre de Oliveira Rocha, Matheus de Freitas Fernandes-Pedrosa, Arnóbio Antônio da Silva-Júnior

**Affiliations:** 1Laboratory of Pharmaceutical Technology and Biotechnology, Department of Pharmacy, Federal University of Rio Grande do Norte (UFRN), Natal-RN 59072-970, Brazil; 2Programa de Pós-Graduação em Bioquímica, Universidade Federal do Rio Grande do Norte, Natal-RN 59072-970, Brazil

**Keywords:** *Bothrops jararaca*, immunoadjuvants, PLGA nanoparticles, nanobiotechnology

## Abstract

Snakebite envenoming represents a worldwide public health issue. Suitable technologies have been investigated for encapsulated recombinant or native proteins capable of inducing an effective and long-lasting adaptive immune response. Nanoparticles are colloidal dispersions that have been used as drug delivery systems for bioactive biological compounds. Venom-loaded nanoparticles modulate the protein release and activate the immune response to produce specific antibodies. In this study, biocompatible cationic nanoparticles with *Bothrops jararaca* venom were prepared to be used as a novel immunoadjuvant that shows a similar or improved immune response in antibody production when compared to a conventional immunoadjuvant (aluminum hydroxide). We prepared stable, small-sized and spherical particles with high Bothrops jararaca venom protein association efficiency. The high protein loading efficiency, electrophoresis, and zeta potential results demonstrated that *Bothrops jararaca* venom is adsorbed on the particle surface, which remained as a stable colloidal dispersion over 6 weeks. The slow protein release occurred and followed parabolic diffusion release kinetics. The in vivo studies demonstrated that venom-loaded nanoparticles were able to produce an immune response similar to that of aluminum hydroxide. The cationic nanoparticles (CNp) as carriers of bioactive molecules, were successfully developed and demonstrated to be a promising immunoadjuvant.

## 1. Introduction

Venomous animal accidents cause about 4 million cases of disability-adjusted life years [1]. In Brazil, about 135,000 cases are registered annually. The genus *Bothrops* stands out as responsible for over 90% of snakebites, where *Bothrops jararaca* is the main species responsible for snakebite accidents [2]. The *Bothrops jararaca* venom is composed of, but not exclusively, metalloproteinases, serine proteinases, L-amino acid oxidases and phospholipase A2. These proteins are related with the development of severe processes of inflammation and aggravation in the victims [3,4]. The treatment for snakebite is immunotherapy with polyspecific antibodies. To improve the immunologic response, immunoadjuvants are associated with antibodies to improve the immunogenicity of antigens. The most employed immunoadjuvants are aluminum salts; however, side effects such as hypersensitivity, granulomatous inflammation, subcutaneous nodules and neurotoxicity have been reported [5,6,7,8]. Research on new immunoadjuvants points to some polymeric biomaterials as promising systems when used at nanometer scales [9,10,11,12].

Nanotechnology applications have expanded gradually over several decades in different areas, including medicine. Drug-loaded nanoparticles have been offered for biomedical applications since 1995 as an alternative to traditional therapies [13]. Even though the safety of the nanoparticles is proven, no standard protocol has used this material for serum production, which remains a challenge for advances in the development of these nanocarriers. Systems of modified drug/biomolecules release are employed to improve therapeutic efficacy and decrease toxic effects [14]. In particular, nanoparticles stand out for their capacity of surface functionalization to improve performance in vitro and in vivo [15]. The functionalization allows cell/tissue-targeting [16,17,18,19], boosts cellular uptake [20,21,22,23] and modulates the immune response [24,25,26].

Some studies have shown a significant interaction between nanomaterials and the ability to stimulate and/or suppress immune responses, and these characteristics have been recently verify with the successful application of nanoparticles to COVID-19 pandemic control [27].

Nanoparticle compatibility outside the immune system is mainly determined by surface properties. The particle size, shape, protein composition and routes of administration are factors contributing to nanoparticle interaction with the immune system [28]. In the context of protein binding to nanocarriers [29,30], nanoparticles can be targeted to modified release of these biomolecules and be used to improve the production of antivenom serum. Passive immunization of animals and the discovery of novel immunomodulators against venomous animals have been extensively studied [31,32,33,34].

The aim of this study was to synthesize and characterize biocompatible functionalized nanoparticles with positive charge for *Bothrops jararaca* venom modified release. On this, we inferred that it is possible to employ nanoparticles to modify the release of venom, providing small amounts during a long exposure in the body to produce a detectable immune response. 

## 2. Results

### 2.1. Cationic PLGA Nanoparticles for B. jararaca Venom-Load Synthesis and Physico-Chemical Characterization

The addition of polyethyleneimine (PEI) polycation to the nanoparticle provides a positive charge to the particle surface, creating cationic nanoparticles (CNp). Figure 1A shows the effect of PEI on the particle size and zeta potential. We used PEI/polymer (*w*/*w*) ratios of 1:5, 1:10, and 1:15, considering the concentration of polymer of 0.75% (*w*/*v*), which statistically increased particle size. However, all of the cationic functionalized formulations presented particle sizes smaller than 150 nm. Figure 1A also shows that cationic zeta potential increased according to the used PEI:PLGA ratios. Thus, the highest 1:5 PEI:PLGA ratio was chosen for further tests, including *B. jararaca* venom entrapment. 

A linear increase in zeta potential, ranging from 0 to 45 mV, was observed with the increase in the PEI/polymer ratio. The average particle diameters also increased from 100 nm to 140 nm with the 1:5 ratio when compared to particles without PEI (Figure 1A). The polydispersity index (PDI) of all systems remained below 0.3, affirming the homogeneity of the particles (Table 1). 

*Bothrops jararaca* venom (BJ) was added onto the CNps in concentrations of 0.5 and 1.0% relative to the polymer weight by adsorption. Table 1 shows the venom loading efficiency (VLE) on the cationic nanoparticles.

Both concentrations of BJ venom had VLE exceeding 98%. This efficiency corroborates with changes in the average particle diameter; venom adsorption produced a slight increase in particle size of 140 nm (venom-free nanoparticles) to 168 nm (venom-loaded nanoparticles). Changes in zeta potential also indicated an apparent disturbance of particle surface charge. CNp particles presented a zeta potential of about 50 mV, and after venom incorporation, the values of zeta potential remained at about 40 mV.

The stability study was performed for 45 days, and results of the average particle diameter can be seen in Figure 1B. Changes in particle size were minimal at the end-point, zeta values remained positive (+30 mV ± 10), and PDI below 0.2 indicated that all systems were homogenous.

The association of BJ proteins and nanoparticles was improved by electrophoresis. The lines with nanoparticles and different concentrations of BJ venom did not present bands, confirming high protein association to systems.

The shape and surface of the CNPs were further assessed by AFM and SEM images. Figure 2 shows spherical particles with smooth surfaces and uniform distribution for all three systems (Figure 2B,C).

### 2.2. Cell Viability Assay

Cell viability was assessed for nanoparticles with PEI 25 kDa in a PBMC assay. Cationic nanoparticles in concentrations of 140, 70, 35, 17.5, 7, and 1.7 μg·mL^−1^ were incubated with blood cells for 72 h.

Cell viability of PLGA nanoparticles with PEI is demonstrated in Figure 3A. After 72 h, most of the concentration was cell compatible; the exception was the treatment with 140 μg·mL^−1^, which presented viability lower than 50%. The effect of nanoparticle concentration on apoptosis is presented in Figure 3B. All concentrations promoted early apoptosis in 25% of cells, and similarly with cell viability, the 140 μg·mL^−1^ treatment promoted late apoptosis. 

### 2.3. Serum Antibody Responses, Evaluation of Antibody Titers and In Vitro Release

The immunization protocol was performed with CNp, CNp + BJ 0.5%, CNp + BJ 1.0%, Al(OH)_3_ + BJ 0.5% and Al(OH)_3_ + BJ 1.0%. Animals treated with aluminum hydroxide or nanoparticles containing venom produced antibodies. Controls groups showed minimal optical density of antibody, confirming the lack of immune response in the absence of venom. High antibodies titers were found in the venom-treated groups; the lowest detectable antibody dilutions were 1 to 51,200. The CNp + BJ groups achieved antibody production equivalent to groups with aluminum hydroxide containing venom, i.e., showed no statistical difference (*p* > 0.05).

The in vitro release assay aimed to observe the influences of system composition on the profile of venom protein release as a function of time (Figure 4C).

The assay was performed for 168 h, equivalent to 7 days. Both systems presented a similarly prolonged release profile. To suggest a kinetic mechanism for release of the protein, mathematical models were applied: the first-order model, Bhaskar model, Freundlich model and parabolic diffusion model. The parabolic diffusion model obtained the highest correlation value when compared with the correlation coefficient (r^2^).

## 3. Discussion

Monitoring nanoparticle parameters is essential for their desired performance. Some studies reported the use of cationic nanoparticles with proteins as safe carriers. These systems are usually responsible for targeting and increasing the effectiveness of biological activity intrinsic to proteins [28,35,36]. PEI is a functionalized agent with high potential to be used as a carrier. PEI molecules anchor and accumulate onto nanoparticle surfaces during the nucleation process, leading to an increase in the diameter as a function of concentration, as observed earlier (Figure 1A). PEI is a 25 kDa polycation with a high cationic density charge that, when organized on nanoparticle surfaces, promotes a positive zeta potential in the diffusion layer (Table 1) [15,37]. The reason for choosing the 1:5 PEI:PLGA ratio was its potential to provide a greater positive charge on the surfaces of the particles after venom-loading. 

The elevated venom load efficiency demonstrated in Table 1 might be a result of charge interaction between BJ venom proteins and the cationic nanoparticles. The importance of charge was also observed in nanoparticle stability. Instability phenomena are related with the interaction of electrostatic, steric and solvation forces; unstable systems might present flocculation, sedimentation and coalescence [38,39,40]. The nanoparticle–PEI behavior was maintained even after association with BJ venom (Figure 1B). Venom proteins possess moderate density of charge (metalloproteases), which helps to maintain the electrostatic and physical stability of the system [41].

The association of proteins with cationic nanoparticles was also identified by gel electrophoresis. The band pattern in Figure 1C shows BJ venom proteins distribution according to their molecular weight. PEI and cationic nanoparticles without venom were used as controls to demonstrate the inability of those compounds to mark in the gel. The empty bands on the lanes of nanoparticles associated with BJ confirmed the elevated association between these molecules [42].

Microscopy images indicated the small (200 nm) and smooth surface of the nanoparticles. These characteristics are important for cellular uptake and desirable biological behavior.

Smooth and spherical particles are not readily internalized by macrophages, and therefore, do not induce strong cytokine secretion [43]. However, the use of highly cationic molecules on the surface of the particle accelerates the process of phagocytosis, improving the interaction with the immune system [28,44].

The highly positive charge density found on PEI is derived from the protonated amines of its monomers. The cationic surfaces of these molecules interact with negative membranes of cells, inducing the endocytosis process of PEI molecules. In the cytosol, free PEI molecules destabilize organelle membranes, leading to cell death, inhibition of lysosomal activity and release of endosomal acids [45,46,47]. This cytotoxic limitation was not found in our study. Cell viability and apoptosis assays showed positive results. 

Thomas et al. (2005) studied the genotoxicity effect of PEI of different molecular weights; as a conclusion, PEI with smaller chains presented a non-toxic profile in DNA, being considered biocompatible [48]. Previously, our research group used poly lactic acid (PLA) nanoparticles and functionalized surface particles with a PEI:PLA ratio of 1:1 and 1:2. After an MTT assay in VERO E6 cells, the test showed a dose-dependent cytotoxicity of PEI in concentrations above 80 µg.mL^−1^ [30]. All of these characterizations contribute to the use of cationic nanoparticles as safe nanocarriers. 

Cationic nanocarriers can be used as immunoadjuvants for improving the signaling immune response in cationic particle uptake. The importance of using immunoadjuvants in passive immunization consists of increasing immune response to antigens that cannot stimulate the immune system by self-properties. Several studies have suggested that PLGA micro- and nanoparticles can be used as immunoadjuvants. This capability can be explained by the fact that PLGA nanoparticles and cationic charges stimulate pro-inflammatory cytokines when phagocytized by macrophages [28,49,50,51]. Although the values did not show statistical difference between the groups, the CNp + BJ 0.5% group had higher dilutions of antibody titers than the other groups. This correlates with the in vitro release assay (Figure 4C), in which the CNp + BJ 0.5% system released almost 100% of its content in a period of 7 days. Thus, CNp + BJ 0.5% has greater efficiency in releasing proteins and inducing the production of antibodies.

CNp slowed the release of venom (Figure 4C). To suggest an adequate kinetic releasing mechanism for the cationic nanoparticles containing *Bothrops jararaca* venom, it was necessary to use different mathematical models such as: first order, Bhaskar, Freundlich and parabolic diffusion (Table 2). The kinetic mechanism for releasing the systems was determined by the analysis of the correlation coefficient (r^2^) obtained through linear regression of the data from application of the mathematical models. The model was chosen when it had an r^2^ closer to 1. In this case, both formulations CNp + BJ 0.5% (r^2^ = 0.96) and CNp + BJ 1.0% (r^2^ = 0.97) suggested that the best representative model was parabolic diffusion, (Mt/M∞)/t = Kp t^−0.5^ + b. This mechanism suggests that the proteins diffuse in the medium through interparticle diffusion or diffusion of proteins adsorbed on the surface of the nanoparticles [30]. In Therefore, venom proteins at the surface of the cationic nanoparticles can be released to tissues in small amounts for days. The cationic surfaces provoke high activation of the immune system and subsequently enhancement of the humoral and cellular response [50]. 

## 4. Conclusions

Produced cationic functionalized PLGA nanoparticles showed suitable physicochemical parameters and physical stability to be applied as colloidal nanocarriers for *Bothrops jararaca* venom release. The experimental results demonstrated an interesting and promising nanoplatform for a cationic PLGA nanoparticle formulation as a biocompatible and efficient immunoadjuvant able to induce serum production and immune response against *Bothrops jararaca* venom.

## 5. Materials and Methods

### 5.1. Animals

BALB/c mice (25–35 g) 6–8 weeks of age were obtained from the “Biotério de Criação de Animais do Centro de Ciências da Saúde da Universidade Federal do Rio Grande do Norte (UFRN, Natal-RN, Brazil)”. Animals were hosted in a controlled temperature room and received food and water ad libitum. The experimental protocol was approved by the “Research Ethics Committee on Animal Use” at the Federal University of Rio Grande do Norte (UFRN) (protocol no. 026.037/2017).

### 5.2. Chemical Reagents

D,L-PLGA 50:50 (lactide:glycolide) (inherent viscosity 0.63 dL·g^−1^ at 30 °C) was purchased from Birmingham Polymers Inc. (Alabaster, AL, USA); poloxamer 407, polyethyleneimine (PEI) branched average molecular weight 25,000 Da and aluminum hydroxide were purchased from Sigma-Aldrich Co. (St. Louis, MO, USA). BCA Protein Assay Kit was purchased from Pierce Biotechnology (Woburn, MA, USA), and Mouse IgG total ELISA Kit from eBioscience (San Diego, CA, USA). The purified water (1.3 µS·cm^−1^) was prepared from reverse osmosis purification equipment (model OS50 LX, Gehaka, São Paulo, Brazil). All other reagents were of analytical grade.

### 5.3. Biological Materials

*Bothrops jararaca* (BJ) snakes were a kind gift from Kathleen Fernandes Grego, Instituto Butantan, São Paulo, Brazil. The venom was obtained by manual extraction from adult specimens and then lyophilized and kept at −20 °C until used. Venom solutions were prepared with PBS at time of use. The amount of venom was expressed by protein content, determined by bicinchoninic acid assay using albumin as standard. The scientific use of the venoms was approved by the Brazilian Genetic Heritage Management Council (CGEN) (Brasilia, Brazil) (Process number 010844/2013-9).

### 5.4. Preparation of Cationic PLGA Nanoparticles for B. jararaca Venom Delivery

Cationic PLGA nanoparticles (CNp) for the incorporation of *B. jararaca* venom were obtained through the nanoprecipitation method previously optimized [21] with some modifications. Briefly, 0.75% *w*/*v* PLGA and distinct ratios of PEI were dissolved in 6 mL of acetone. The PLGA + PEI solutions were then injected at an outflux of 1.0 mL·min^−1^ into 14 mL of aqueous solution of poloxamer 407 0.5% *w*/*v*, under magnetic stirring. The effect of PEI 25 kDa concentration on the nanoparticle formation was investigated using various ratios of PEI:PLGA (1:5, 1:10, 1:15 *w*/*w*). The samples remained under magnetic stirring at 25 °C ± 2 for 24 h for solvent evaporation.

For protein loading in cationic PLGA nanoparticles, different ratios of *B. jararaca* venom (0.5 and 1.0%—weight relative to PLGA concentration) were dissolved previously in the PBS, which was maintained at a temperature of 20 ± 2 °C before the nanoparticle preparation procedure. Under magnetic stirring, PLGA nanoparticles and BJ venom were mixed for 1 h in a closed vial flask. 

### 5.5. Venom Loading Efficiency 

The different BJ venom-loaded PLGA nanoparticles were centrifuged at 20,000× *g* at 4 °C for 30 min, and the supernatant was removed and subjected to protein analysis using the BCA Protein Assay Kit through the bicinchoninic acid method. The assays were performed in triplicate, and the venom loading efficiency (VLE%) was calculated using Equation (1):VLE% = [(total amount protein − free amount protein in supernatant)/total amount protein] × 100(1)

### 5.6. Electrophoresis

The electrophoretic profile of *B. jararaca* venom, PEI, venom-free cationic nanoparticles, and distinct protein-loaded nanoparticles were determined by polyacrylamide (15%) gel electrophoresis and sodium dodecyl sulfate (SDS-PAGE), using an electrophoresis cell (Mini-ProteanTM II; Biorad, Hercules, CA, USA) [13]. All of the samples were prepared with equivalent weights relative to PLGA concentration 1% of 0.75% *w*/*v* (15 µg). The relative molecular mass of proteins was monitored and compared with the electrophoretic migration pattern of a standard protein mixture (Gibco-BRL Life Technologies, Gaithersburg, MD, USA). All of the samples were processed under reduction conditions with 10% β-mercaptoethanol (sample ratio 1:1). The gels were stained by the Coomassie stain, 0.1% silver nitrate protocol [14], and then scanned.

### 5.7. Physicochemical Aspects and Colloidal Stability

The measurements of mean diameter and particle size distribution were performed by dynamic light scattering in a ZetaPlus (Brookhaven Instruments Co., Holtsville, NY, USA), equipped with a 90Plus/BI-MAS apparatus, at a wavelength of 659 nm with a scattering angle of 90°. The zeta potential of the particles was measured by laser Doppler anemometry using the same equipment. All analyses were performed at 25 °C. Experimental values are given as the mean ± SD for the experiments carried out in triplicate for each sample. 

The shape and topographical images of the samples were observed on atomic force microscopy (AFM) images (SPM-9700, Shimadzu, Tokyo, Japan) using a silicon tip probe (Nanoworld, Neuchatel, Switzerland,). The dispersions were freshly diluted with purified water at a ratio of 1:25 (*v*/*v*), dropped on a cover slip, dried under a desiccator for 24 h, and then analyzed using an AFM (SPM-9700, Shimadzu, Tokyo, Japan) at room temperature by scanning with a cantilever (non-contact) at 1 Hz.

The physical stability study was performed with cationic nanoparticles with or without BJ venom. Analysis was performed for 45 days, evaluating the mean size, zeta potential and polydispersity index. The particles were stored in a freezer at 5 °C ± 2 °C; before analysis, the samples were stabilized at 25 °C ± 2 °C.

### 5.8. Release Profile Study In Vitro

The protein release profile from the cationic nanoparticles was monitored in 1 mL of buffered solution (pH = 7.4, KH_2_PO_4_ 0.05 M) in a thermostatic bath (model SL-150/22, Solab) at 37 °C ± 0.2 °C. For 7 days, at specific intervals, the flasks were centrifuged at 16,000× *g* for 30 min, the supernatant removed, and the protein content determined using the BCA Protein Assay Kit through the bicinchoninic acid method, in triplicate. The cumulative percentage of released protein was plotted versus time and fitted using different mathematical models: first order, Bhaskar, Freundlich and parabolic diffusion model.

### 5.9. Cell Viability and Apoptosis Assays

Heparinized blood samples were collected from all recruited individuals. The survey was first approved by the Research with Human Beings Ethics Committee of the Federal University of Rio Grande do Norte (protocol CAAE: 56191416.0.0000.5537). PBMCs were separated by centrifugation over a gradient of Ficoll-Hypaque (Merk, São Paulo, Brazil). Mononuclear cells were resuspended in RPMI supplemented with 10% human AB Rh-serum (Merk, São Paulo, Brazil), 10 mM HEPES (Merk, São Paulo, Brazil), 1.5 µM L-glutamine (Merk, São Paulo, Brazil), 200 IU of penicillin per ml, and 200 µg of streptomycin per ml (Merk, São Paulo, Brazil) and were adjusted to 2 × 10^6^ cells/mL. PBMCs (10^6^ cells per well) were cultured in vitro in 24-well flat bottom plates (Nunc, Roskilde, Denmark) at 37 °C in a humidified atmosphere of 5% CO_2_ in air and in the presence of different concentrations of blank-cationic nanoparticles or medium alone. After 3 days in culture, the stimulated cells were harvested and processed for apoptosis analyses. For apoptosis analyses, FITC Annexin V apoptosis detection kits (BD) were used following the manufacturer’s instructions. Twenty thousand events were acquired from each sample into a lymphocyte gate using a FACS Canto II flow cytometer (BD, Canto II flow cytometer, Becton Dickinson Bioscience, San Jose, CA, USA). Annexin V and Pi were analyzed using the FlowJo vX software.

### 5.10. Vaccinations

Experimental mice were immunized for 6 weeks with 100 µL of subcutaneous injections administered bilaterally in the lumbar region with BJ venom in different concentrations (0.5 or 1.0%), adsorbed in cationic PLGA nanoparticles or associated with aluminum hydroxide.

### 5.11. Serum Production 

The experimental mice were bled by cardiac puncture. Blood samples in the absence of an anticoagulant were incubated at 37 °C for 30 min and then at 4 °C for 2 h for clot retraction. Then, the samples were centrifuged at 15,000× *g* for 5 min at 4 °C, and the supernatant (serum) collected and stored at −20 °C.

### 5.12. Serum Antibody Responses 

Antigen-specific serum antibody responses were measured 1 week following the booster hyperimmunization by ELISA. The ELISA assay was performed following the protocol of eBioscience. The plate was sensitized with 100 µL/well of capture antibody in coating buffer and sealed and incubated overnight at 4 °C. After this step, the wells were washed twice with 400 µL of wash buffer solution. The wells were blocked with 250 µL of blocking buffer and incubated at room temperature for 2 h. After washing the plate again, two-fold serial dilutions of the standards were performed with assay buffer to produce the standard curve. For each well, 100 µL of assay buffer were added to the blank wells and 90 µL added to the sample wells; after this, 10 µL/well of pre-diluted samples were added in assay buffer to the appropriate wells, and 50 µL/well of diluted detection-antibody was added to all wells. The plate was sealed and incubated at room temperature for 3 h. After washing, substrate was added, and the plate incubated at room temperature for 15 min. The reaction was stopped, and the plate read at 450 nm.

### 5.13. Statistical Analysis

Data were expressed as the mean ± standard deviation (SD). Statistical significance of differences was assessed by Student’s *t*-test or ANOVA, followed by Tukey’s multiple comparison. A probability value of *p* < 0.05 (*) was considered significant, and *p* < 0.01 (**) was considered very significant.

## Figures and Tables

**Figure 1 toxins-14-00888-f001:**
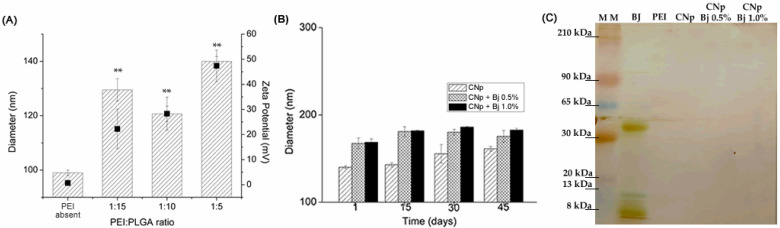
(**A**) Mean diameter and zeta potential of PLGA nanoparticles as function of PEI: PLGA ratio variation; ** *p* < 0.01 for the size of cationic group compared to the PEI-absent group. (**B**) Physical stability study of cationic nanoparticles (CNps); cationic nanoparticles with *Bothrops jararaca* venom 0.5% (CNp + BJ 0.5%); cationic nanoparticles with *Bothrops jararaca* venom 1.0% (CNp + BJ 1.0%), over 45 days; ** *p* < 0.01 and compared to the size of the CNps on day 1. (**C**) Electrophoretic profile of: MM, molecular marker; BJ, *Bothrops jararaca* venom; PEI, polyethyleneimine; CNp, cationic nanoparticles. Gel stained with Coomassie stain protocol.

**Figure 2 toxins-14-00888-f002:**
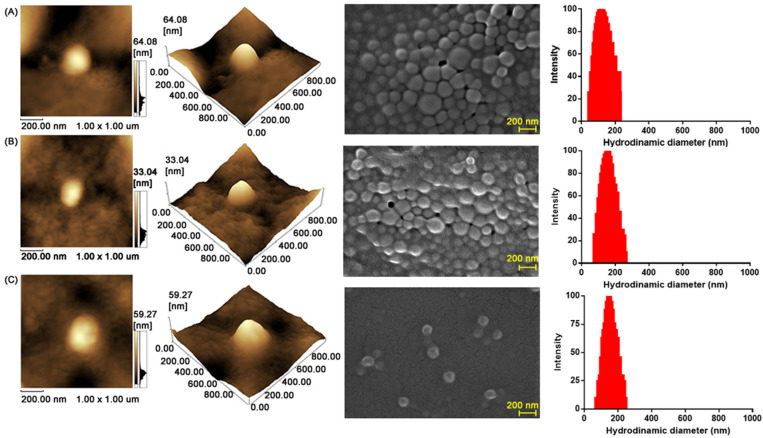
2D and 3D topographic images obtained by AFM, SEM images and distribution of particles, respectively. (**A**) venom-free cationic nanoparticles; (**B**) PLGA cationic nanoparticles with 0.5% *Bothrops jararaca* venom; and (**C**) PLGA cationic nanoparticles with 1.0% *Bothrops jararaca* venom.

**Figure 3 toxins-14-00888-f003:**
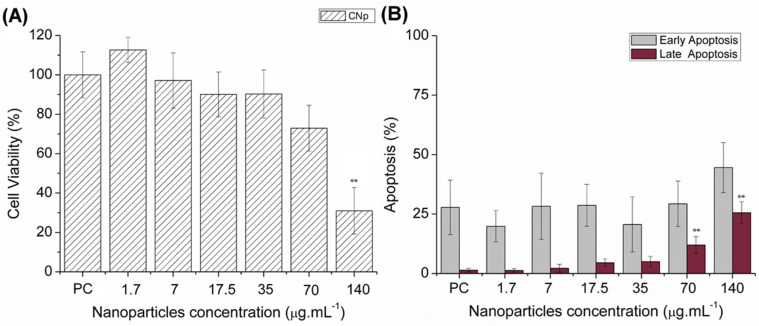
Cell viability (**A**) and cellular apoptosis rate (**B**) in blood cells using cationic nanoparticles in different concentrations (1.7, 7, 17.5, 35, 70 and 140 μg·mL^−1^) for 72 h. Note: CNp—Cationic nanoparticles; PC—Positive control; ** *p* < 0.01 and for the nanoparticle concentration compared to the positive control.

**Figure 4 toxins-14-00888-f004:**
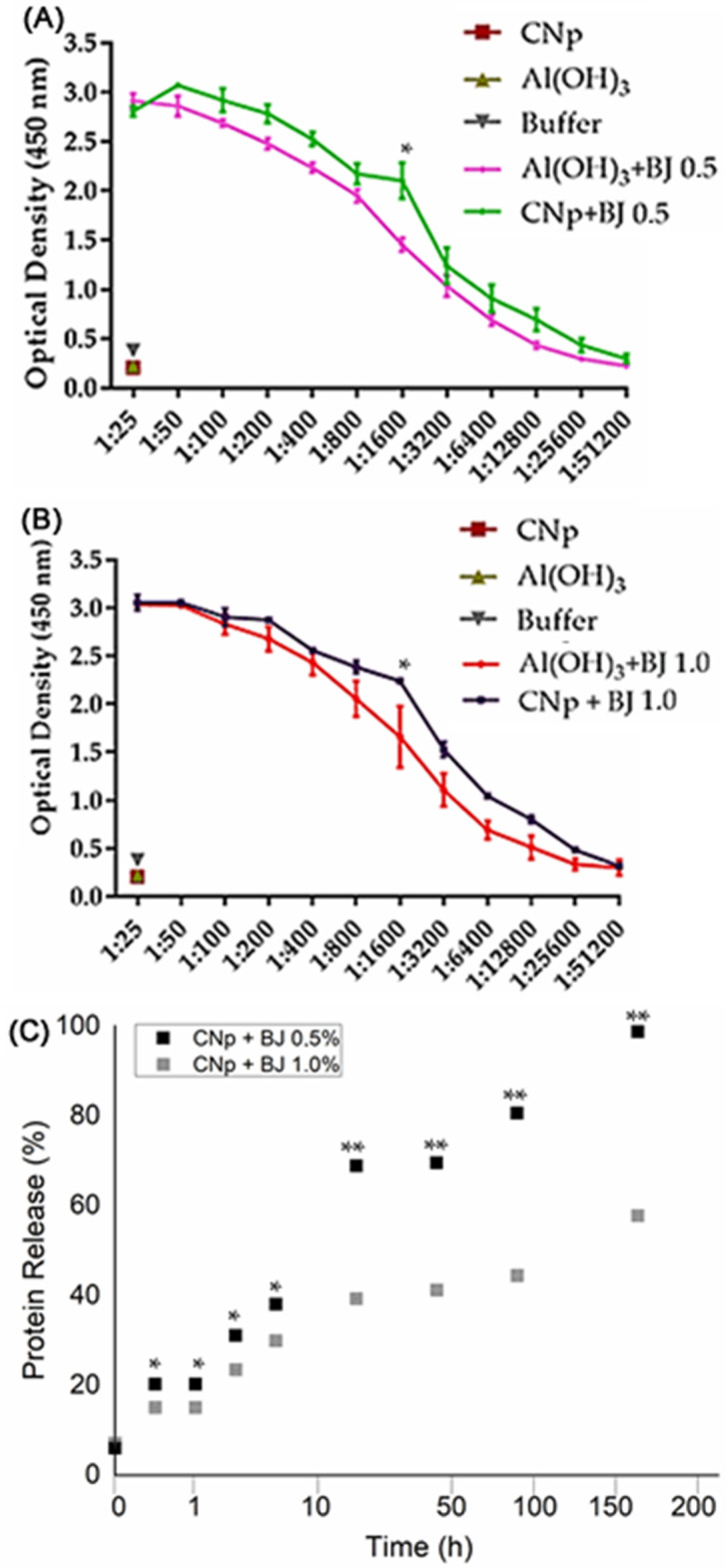
Evaluation of antibodies obtained from animals immunized with aluminum hydroxide (Al(OH)_3_) and cationic nanoparticles (CNp) associated with *Bothrops jararaca* venom (BJ) at different concentrations of (**A**) 0.5% and (**B**) 1.0%; ** *p* < 0.01 and * *p* < 0.05 for the CNp compared to the aluminum hydroxide. (**C**) In vitro protein release profile of venom at different concentrations (0.5 and 1.0%) associated with cationic nanoparticles, as a function of time; ** *p* < 0.01 and * *p* < 0.05 for the CNp + BJ 0.5% compared to the CNp + BJ 1.0%.

**Table 1 toxins-14-00888-t001:** Venom loading efficiency and physicochemical parameters of different NP formulations.

Systems	Particle Size (nm)	PDI	Zeta Potential (mV)	VLE (%)
CNp	140.0 ± 1.8	0.13 ± 0.01	47.38 ± 6.24	-
CNp + 0.5% BJ	167.3 ± 6.3 *	0.09 ± 0.02	42.83 ± 14.6	98.7 ± 0.07 *
CNp + 1.0% BJ	168.7 ± 3.8 *	0.09 ± 0.01	35.6 ± 13.4	98.2 ± 0.03 *

Notes: PLGA, poly (lactic-co-glycolic acid); CNp, cationic nanoparticles; PDI, polydispersity index; BJ, *Bothrops jararaca* venom; VLE, venom loading efficiency. The results are expressed as mean ± SD (n = 3) * *p* < 0.01 and for the venom group compared to the CNp group.

**Table 2 toxins-14-00888-t002:** Mathematical treatment for in vitro protein release study.

		Kinetic	Model *k (r^2^)*	
Systems	First Order	Bhaskar	Freundlich	Parabolic Diffusion
CNp + BJ 0.5%	0.0003 h^−1^ (0.59 ± 0.03)	0.1741 h^0.65^ (0.90 ± 0.02)	145.11 h (0.91 ± 0.02)	0.2722 h^−0.5^ (0.96 ± 0.01)
CNp + BJ 1.0%	0.0002 h^−1^ (0.56 ± 0.04)	0.1329 h^0.65^ (0.89 ± 0.02)	89.62 h (0.94 ± 0.01)	0.3635 h^−0.5^ (0.97 ± 0.01)

Notes: CNp, cationic nanoparticles; BJ, *Bothrops jararaca* venom; The results are expressed as mean ± SD (n = 3).

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
