# Peer review of "Cationic PLGA Nanoparticle Formulations as Biocompatible Immunoadjuvant for Serum Production and Immune Response against Bothrops jararaca Venom"

_toxins, 2022, doi:10.3390/toxins14120888_

Round 1

Reviewer 1 Report

The manuscript is dedicated to the synthesis, characterization of PLGA/PEI nanoparticles for the immunoadjuvants. A lot of drawbacks in the manuscript do not allow to accept it in current form.

1.Introduction. The general idea of the research is unclear. Do authors suggest to make a vaccine for immunization toward snake bites? It sounds non-realistic. As a treatment of poisoned people the suggested nanoparticles are not applicable.

2.” venom is composed mainly of toxic proteins, already known as: metalloproteinases, serine proteinases, L-amino acid oxidases and phospholipases A2” – phospholipase A2 is not toxic protein. 

3. Figure 1A. The label 1:0 for PEI:PLGA ratio means that you have pure PEI while the text reflects reverse situation.

4.” An interesting data to be noted was the distribution of average diameters of systems containing venom which showed polydispersity values less than 0.010.” The Figure 2 does not reflect such narrow distribution. Also, it is unclear does protein adsorb on the surface of particles or incorporated in them. In first case the low PDI is instrumentational error.

5.” The shape and surface of free-venom and venom-loaded CNps were further accessed by AFM and SEM images.” AFM images were not processed properly.

6. Control experiments with PEI are absent

7.The role of the nanoparticles/PEI complex remains uncertain in discussion part. Does this complexation just lowers number of active PEI units, how the complex PLGA/PEI is organized? What is the role of PLGA carrier?

8.The characteristics of the initial PLGA are not full, so it is not easy to analyze the results. The quantitative analysis of the complex is not presented. The experimental part of synthesis is shortened and hardly could be reproduced.

Author Response

RESPONSE TO THE COMMENTS OF THE REVIEWER #1

General comment:

- The manuscript is dedicated to the synthesis, characterization of PLGA/PEI nanoparticles for the immunoadjuvants. A lot of drawbacks in the manuscript do not allow to accept it in current form.:

Response: We appreciate the referee’s comment. We believe that all the suggestions contributed to improving the clarity and quality of the message in the manuscript. We revised the entire manuscript to improve the explanation about our proposal according to the suggestions. Please find below the responses to the comments and an itemized list of changes in the manuscript highlighted in RED.

Specific comment 1: Introduction. The general idea of the research is unclear. Do authors suggest to make a

vaccine for immunization toward snake bites? It sounds non-realistic. As a treatment of poisoned

people the suggested nanoparticles are not applicable.

Response:

Thank you for bringing this point to our attention and for signaling the need to improve our proposal explanation. The proposal of the study is the synthesis and characterization of biocompatible cationic nanoparticles loaded with Bothrops jararaca venom (Bj) to be used as a novel immunoadjuvant that shows a similar or improved immune response in antibody production when compared to a conventional immunoadjuvant (aluminum hydroxide). For this proposal, animals were immunized for 6 weeks with Bj venom at different concentrations (0.5 or 1.0%), encapsulated in cationic nanoparticles or associated with aluminum hydroxide. After the last immunization, the animals were euthanized and serum was collected for IgG measurement. Our results showed that the venom-associated nanocarrier demonstrated efficient IgG production, indicating that the formulation could be used as an alternative immunoadjuvant for serum production against Bj snake venom. The introduction has been carefully reworked to clarify our idea. Please check lines 36-75.

Specific comment 2: “venom is composed mainly of toxic proteins, already known as: metalloproteinases, serine proteinases, L-amino acid oxidases and phospholipases A2” – phospholipase A2 is not toxic protein.

Response: To improve clarity the description, we decided to rephrase the sentence to “The Bothrops jararaca venom is composed by, but not exclusively, metalloproteinases, serine proteinases, L-amino acid oxidases and phospholipases A2. These proteins are related with the development of severe processes of inflammation and aggravation of victim”. Please also check the lines 39-42.

Specific comment 3: Figure 1A. The label 1:0 for PEI:PLGA ratio means that you have pure PEI while the text reflects reverse situation.

Response: To improve clarity to figure 1A, we decided to re-edit the figure. In fact, the first point of “Mean diameter and Zeta Potential of PLGA nanoparticles” is refers to the system without PEI. Please check the page 2, figure 1A.

Specific comment 4: “An interesting data to be noted was the distribution of average diameters of systems containing venom which showed polydispersity values less than 0.010.” The Figure 2 does not reflect such narrow distribution. Also, it is unclear does protein adsorb on the surface of particles or incorporated in them. In first case the low PDI is instrumentational error.

Response: This remark is completely appropriate. After analyzing data that gave rise to the particle distribution figure, we found the error. Table 1 has been properly corrected. Please check correct polydispersity index (Table 1). To clarify the surface characteristics, section “5.4 Preparation cationic PLGA nanoparticles for B. jararaca venom delivery” was properly edited.

Specific comment 5: “The shape and surface of free-venom and venom-loaded CNps were further accessed by AFM and SEM images.” AFM images were not processed properly.

Response: We have considered the referee’s concern previously because some studies have applied and discussed AFM images to give information about the mean size, shape, and surface of nanoparticles. As for example: (C.M. Hoo, N. Starostin, P. West, M.L. Mecartney, A comparison of atomic force microscopy (AFM) and dynamic light scattering (DLS) methods to characterize nanoparticle size distributions, J. Nanoparticle Res. 10 (2008) 89–96; H.S. Kim, W.I. Park, M. Kang, H.J. Jin, Multiple light scattering measurement and stability analysis of aqueous carbon nanotube dispersions, J. Phys. Chem. Solids. 69 (2008) 1209–1212. doi:10.1016/j.jpcs.2007.10.062). However, about this issue we have considered only the DLS measurements because we have adjusted the AFM equipment for assessing only the shape and surface aspects of nanoparticles. For size comparisons, a greater number of images and particles should be considered. In addition, the AFM image does not consider the hydrodynamic radius of nanoparticles. The preparation of samples for analysis can leads to alteration of size of particles due to the aqueous phase evaporation.

Specific comment 6: Control experiments with PEI are absent

Response: PEI is a well-known polycation commonly used for gene transfection. Scientific evidence extensively shows the high toxicity capacity of the free molecule in cells and tissues. If that specific comment was about In vivo experiments, a PEI control group could show interesting results as a vehicle to venom protein. However, the probability to cause tissue toxicity is high, changing the perspective of be using as a safe vehicle. If that specific comment was about cell viability, the result itself shows toxicity nanoparticles concentration dependent and others previous studies in our group already published the in vitro PEI toxicity. Due this points, PEI is considered as compound present in nanoparticle formulation and not as isolated nanocarrier. Thus, we have used PEI-functionalized NPs and PEI absent NPs in our experiments.

Specific comment 7: The role of the nanoparticles/PEI complex remains uncertain in discussion part. Does this complexation just lowers number of active PEI units, how the complex PLGA/PEI is organized? What is the role of PLGA carrier?

Response: We understand the referee’s concern about the interactions among the referred molecules and the mechanisms involved. Some previous studies reported in the literature worry about to understand the protein interactions with different polymers, using experimental spectroscopy and theoretical approaches. However, our study was performed in a different way. We looked for understand how the composition of these particles modulates the physicochemical properties, the biocompatibility, the stability, the protein loading, and kinetic release properties. Nanoprecipitation method consists in a hydrophobic polymer (PLGA) precipitation. Previous studies in our group, optimized this method and we were successfully able to synthesize stable nanoparticle based-aliphatic polyester as the PLGA-based. We have established this using a simple and reproducible way for a low-energy nanoprecipitation method for further application in the biomedical and biotechnological purposes. The fabrication protocol for the PLGA-PEI nanoparticles is based on a previously developed formulation, which was optimized and published in a previous study. The composition contains PLGA 0.75% (w/v), polymeric surfactant poloxamer 407 0.5% (w/v) and functionalizing agent PEI 0.075%. All raw materials are certified and obtained from qualified suppliers. The concentrations of the components of empty nanoparticle formulation are not verified by any quantitative tests. The PLGA nanoparticle is the main nanocarrier, which is decorated with PEI to improve interaction with negatively charged compounds, such as proteins and cell surface.

Specific comment 8: The characteristics of the initial PLGA are not full, so it is not easy to analyze the results. The quantitative analysis of the complex is not presented. The experimental part of synthesis is shortened and hardly could be reproduced.

Response: This remark is completely appropriate. By nanoprecipitation method, the nanoparticles synthesis is performed for self-assembled after solvent-displacement and consequent PLGA co-precipitation. To improve clarity of the experimental part of synthesis, section “5.4 Preparation cationic PLGA nanoparticles for B. jararaca venom delivery” was properly edited. The quantitative analysis of the complex is not presented, because the experimental approach it is based in a simple and reproducible way for a low-energy nanoprecipitation method.

Reviewer 2 Report

Cationic PLGA nanoparticle formulations as biocompatible immunoadjuvant for serum production and immune response against Bothrops jararaca venom. Submitted to section: Animal Venoms.   Below, my suggestions are presented in blue.

Lane 7: Complete de idea: proteins capable of inducing antibody production to response …..

Lane 10: to change.  Nanoparticles

Lane 11: to change bioactive macromolecules for bioactive biological compounds, because macromolecules are big proteins, DNA lipid. In fact, peptide, DNA fragment or chemical compounds are used.

Lane 12: to change and to characterize for:  characterization of biocompatible cationic nanoparticles with Bothrops jararaca venom.

Lane 13-14: to change for: Show a stable small-sized and spherical particles with protein association efficiency of about 100% of…? Added a; complete the sentences.

Lane 17:  To change: The in vivo studies demonstrated that venom-loaded nanoparticles were able to produce a similar immune response compared with aluminum hydroxide, an adjuvant widely used in immunization protocol.

Lane 19: To change: The cationic nanoparticles (CNp) as carrier to bioactive molecules was successfully developed and demonstrated to be a promising immunoadjuvant and a novel nanocarrier for protein or nucleic acids release.

Lane 30: To change for: with to for

Lane 31: The sentences  “The genus Bothrops is responsible for over 90% of snakebites and Bothrops jararaca species are responsible for leading cause of death  or disability-adjusted life years of survivors” must be improved

Lane 36: To change: Proteins are related with severe processes of inflammation and aggravation of victims..

Lane 41  ¿It is correct the word  perception or response?

Lane 50: I believe that is a better the sentence as following: The nanotechnology application has expanded gradually for several decades in different areas as medicine.

LANE 50-51. The sentences “Through the Pharmaceutical Industry, nanoparticle products are already offered to biomedical application since 1995 as an alternative to traditional therapies”

It is not clear the sentences, because nanoparticle products are not used in the pharmaceutical industry, if not, drugs or molecules coupled to nanoparticles to improve their release

Lane 57. a reference is missing

Lane 81 The addition of PEI polycation in the produce process will give a positive charge on surface particle creating cationic nanoparticles (CNp)

Lane  91  : The data show a linear increase of the zeta potential value

Finally, in the text they are several grammatical mistakes; also, could be interesting to discuss some aspects indicated in the reference by Biswas A, et al 2012 PMID: 23236583;

Some words are in italic and the apoptosis plots or histograms using annexin -V should be included in the manuscripts

Author Response

RESPONSE TO THE COMMENTS OF THE REVIEWER #2

General comment:

- Cationic PLGA nanoparticle formulations as biocompatible immunoadjuvant for serum production and immune response against Bothrops jararaca venom. Submitted to section: Animal Venoms. Below, my suggestions are presented in blue.

Response: We appreciate the referee’s comment. We believe that all the suggestions contributed to improving the clarity and quality of the message in the manuscript. Please find the responses and altered text highlighted in RED.

Specific comment 1: Complete the idea: proteins capable of inducing antibody production to response …

Response: To improve clarity the description, we decided to rephrase the sentence to “Suitable technologies have been investigated for encapsulated recombinant or native proteins capable of induce effective and long-lasting adaptative immune response.” Please also check the lines 15-16.

Specific comment 2: The sentences “The genus Bothrops is responsible for over 90% of snakebites and Bothrops jararaca species are responsible for leading cause of death or disability-adjusted life years of survivors” must be improved

Response: To improve clarity the description, we decided to rephrase the sentence to “The genus Bothrops stands out as responsible for over 90% of snakebites, where Bothrops jararaca species are the main responsible for snakebite accident.” Please also check the lines 37-39.

Specific comment 3: The sentences “Through the Pharmaceutical Industry, nanoparticle products are already offered to biomedical application since 1995 as an alternative to traditional therapies” It is not clear the sentences, because nanoparticle products are not used in the pharmaceutical industry, if not, drugs or molecules coupled to nanoparticles to improve their release.

Response: To improve clarity the description, we decided to rephrase the sentence to “The nanotechnology application has expanded gradually for several decades in different areas as medicine. Drug-loaded nanoparticles are already offered to biomedical application since 1995 as an alternative to traditional therapies.” Please also check the lines 59-61.

Specific comment 4: Finally, in the text they are several grammatical mistakes; also, could be interesting to discuss some aspects indicated in the reference by Biswas A, et al 2012 PMID: 23236583;

Response: We have consulted and cited adequately the referred reference. This study helped us to improve the introduction. The revised version of the manuscript has been edited for spelling and grammar.

Specific comment 5: Some words are in italic and the apoptosis plots or histograms using annexin -V should be included in the manuscripts

Response: To improve clarity to Figure 3, we decided to re-edit the figure. In fact, some words in italic to distinguish snake specie name. Please check Figure 3, page 5.

Reviewer 3 Report

The developed methods are adequate to the tasks set. but the question remains about the toxicity of the resulting drug delivery vehicle. In the future, it is necessary to evaluate the toxicological potential of the developed cationic nanoparticles, since the basis of nanoparticles are animal venoms. With increasing dosage, the toxic load on the body will depend on the carrier, and not on the active substance.

Serum antibody responses - in the text of the method and the results of the study, the validation of the method is not indicated and the reliability of obtaining results by the ELISA method is not clear. Researchers should explain how validation is carried out and how reliable results are obtained.

Supplement the comparative characteristics in the text of the results of 05 and 1% concentrations in each method. Show the benefits of one of them.

Author Response

RESPONSE TO THE COMMENTS OF THE REVIEWER #3

General comment:

- The developed methods are adequate to the tasks set. but the question remains about the toxicity of the resulting drug delivery vehicle. In the future, it is necessary to evaluate the toxicological potential of the developed cationic nanoparticles since the basis of nanoparticles are animal venoms. With increasing dosage, the toxic load on the body will depend on the carrier, and not on the active substance.

Response: We appreciate the referee’s comment. We believe that all the suggestions contributed to improving the clarity and quality of the message in the manuscript. Please find the responses and altered text highlighted in RED.

Specific comment 1: Serum antibody responses - in the text of the method and the results of the study, the validation of the method is not indicated and the reliability of obtaining results by the ELISA method is not clear. Researchers should explain how validation is carried out and how reliable results are obtained.

Response: The antibody quantification analysis and the obtaining of antibody titers follow the eBioscience quantification protocol, which is previously validated by the company's guarantee. Some units of measurement were in altered magnitudes, which have been corrected. Please check the section 5.12 "Serum antibody responses", lines 368-380.

Specific comment 2: Supplement the comparative characteristics in the text of the results of 0.5 and 1%

concentrations in each method. Show the benefits of one of them.

Response: After analyzing the results, we observed that formulation containing 0.5% BJ venom had the best performance.  This is confirmed by the fact that antibody titer of 0.5% BJ venom formulation allows for more dilutions than the 1% BJ venom formulation, which may be associated with its release profile that releases the venom completely within the assay period (7 days). To improve clarity, we decided to rewrite the discussion about both systems. Please the lines 225-249.

Reviewer 4 Report

The manuscript entitled “ Cationic PLGA nanoparticle formulations as biocompatible immunoadjuvant for serum production and immune response against Bothrops jararaca venom by unknown authors talks about the Characterization of biocompatible cationic nanoparticles to Bothrops jararaca venom loading.

Although the results are promising and potentially applicable, there are serious flaws in the writing of the manuscript. It is hard to follow at times. The information is not properly delivered by the authors.

I would request the authors to get the whole manuscript re-written by a native English speaker and resubmit it to the journal. The manuscript is unacceptable in the present form.

Specific examples of English flaws below:

Line 13-14: If ita 100% ,then why “about”

Line 29, In Brazil instead of At Brazil

Line 41: for increase the perception of the pathogen.

Line 73: On 73 this, it can be inferring 

Line 58: can be targeted instead of can be target

Line 72: “study is to obtaining” is incorrect

Line 82: creating

Line 405: A probability value (p) of < 0.05 was considered significant, and p < 0.01 was 405 considered very significant. I do not see any asterisks or star on the graphs

Line 241: Thomas et al. instead of Thomas and colleagues.

It's unfortunate that the manuscript fails to deliver the message mostly due to a lack of writing process.

Author Response

RESPONSE TO THE COMMENTS OF THE REVIEWER #4

General comment:

- The manuscript entitled “Cationic PLGA nanoparticle formulations as biocompatible immunoadjuvant for serum production and immune response against Bothrops jararaca venom by unknown authors talks about the Characterization of biocompatible cationic nanoparticles to Bothrops jararaca venom loading.

Although the results are promising and potentially applicable, there are serious flaws in the writing of the manuscript. It is hard to follow at times. The information is not properly delivered by the authors.

I would request the authors to get the whole manuscript re-written by a native English speaker and resubmit it to the journal. The manuscript is unacceptable in the present form.

It's unfortunate that the manuscript fails to deliver the message mostly due to a lack of writing process.

Response: We appreciate the referee’s comment. We believe that all the suggestions contributed to improving the clarity and quality of the message in the manuscript. We did a thorough job of re-evaluating the text to improve the understanding of our results. Please, find the responses and altered text in RED.

Specific comment 1: A probability value (p) of < 0.05 was considered significant, and p < 0.01 was considered very significant. I do not see any asterisks or star on the graphs

Response: In fact, the statistical treatment went unnoticed. Figure 1, 3, 4 and Table 1 was re-edited and notes have been added to each of the elements. Please check the pages 3-6.

Reviewer 5 Report

I have carefully read the article entitled “Cationic PLGA nanoparticle formulations as biocompatible immunoadjuvant for serum production and immune response against Bothrops jararaca venom” in which authors obtained and characterized biodegradable and biocompatible functionalized nanoparticles with positive charge for releasing of Bothrops jararaca modified venom, are useful to produce measurable immunogenic response in immunization system in vivo model. The introduction is well written, however I suggest a few revisions of the writing style of the manuscript. Moreover, the few results performed, they are in general well presented expect for gel electrophoresis and the lack of statical analysis. Thus, I suggest to revise the manuscript before possible publication in “Toxins” journal.

In particular, the gel electrophoresis in Figure 1C. It is an SDS-PAGE analysis? Which acrylamide percentage? In presence or absence of reducing agent? Which protein amount? Please add all this information in Figure legend, including the meaning of acronyms. Some information need to be included also in the materials and methods. Moreover, most of markers are not visible. I suggest to repeat the electrophoresis.

Figure 3. Check the numbers. Sometimes authors used coma instead of dot. In the Figure legend, change “μg.mL-1” by “µg mL-1”.

Page 5, line 137: change “KDa” by “kDa”. Check in overall text;

Page 5, line 140: change “μg / mL” by “μg mL-1” and “hours” by “h”. Check in overall text;

Page 5, lines 140-142: . Please use always “Figure”. Check in overall text;

Page 6, line 150: change “μg / mL” by “μg mL-1” and “hrs” by “h”. Check in overall text;

Page 6, line 160: please che the subscript in“Al(OH)3”;

Figure 4. Check the numbers. Sometimes authors used coma instead of dot. Please check this in all Figures;

Page 9, line 270: Sometimes authors used coma instead of dot. Please check this in overall manuscript;

In general, the Figure captions need to be revised, adding some information more (e.g. assay used for apoptosis evaluation and so on);

Try to increase the images resolution;

In general, in the methods section authors have to check numbers and measure unit, the use of apex and subscripts. Change “cells/ml” by “cells/mL” or “mol.L−1” by “M” (line 355).

Authors include the paragraph for statistical analysis, but they don’t show this in any Figures. Please revise all Figures, including statistical analysis.

Author Response

RESPONSE TO THE COMMENTS OF THE REVIEWER #5

General comment:

- I have carefully read the article entitled “Cationic PLGA nanoparticle formulations as
biocompatible immunoadjuvant for serum production and immune response against Bothrops
jararaca
venom” in which authors obtained and characterized biodegradable and biocompatible
functionalized nanoparticles with positive charge for releasing of Bothrops jararaca modified
venom, are useful to produce measurable immunogenic response in immunization system in
vivo
model. The introduction is well written, however I suggest a few revisions of the writing style
of the manuscript. Moreover, the few results performed, they are in general well presented
expect for gel electrophoresis and the lack of statical analysis. Thus, I suggest to revise the
manuscript before possible publication in “Toxins” journal.

Response: We appreciate the referee’s comment. We believe that all the suggestions contributed to improving the clarity and quality of the message in the manuscript. We revised the entire manuscript to improve the explanation about our proposal according to the suggestions. Please find the responses and altered text in RED.

Specific comment 1: In particular, the gel electrophoresis in Figure 1C. It is an SDS-PAGE analysis? Which acrylamide percentage? In presence or absence of reducing agent? Which protein amount? Please add all this information in Figure legend, including the meaning of acronyms. Some information need to be included also in the materials and methods. Moreover, most of markers are not visible. I suggest to repeat the electrophoresis.

Response: The electrophoretic profile was determined by polyacrylamide (15%) gel electrophoresis and sodium dodecyl sulfate (SDS-PAGE). All the samples were performed under reduction conditions with 10% β-mercaptoethanol (sample ratio 1:1). All the samples were performed with equivalent amount of weight relative to PLGA concentration, 1% of 0.75% w/v of polymer (15 µg). Figure 1C was re-edited to improve clarity and the figure legend was rewritten. Also, the section 5.6 “Electrophoresis” was checked and improved. Please check the section.

Specific comment 2: In general, the Figure captions need to be revised, adding some information more (e.g. assay used for apoptosis evaluation and so on). Authors include the paragraph for statistical analysis, but they don’t show this in any Figures. Please revise all Figures, including statistical analysis.

Response: In fact, the statistical treatment went unnoticed. We improve the section 5.13 “Statistical Analysis” to apply at results. The probability value (p) of < 0.05 considered significant, and p < 0.01 considered very significant. Figure 1, 3, 4 and Table 1 was re-edited and notes have been added to each of the elements. Please check the pages 3-6. All figures are in good resolution and in .tiff format. We may attach separately from the manuscript for further evaluation.

Specific comment 3: In general, in the methods section authors have to check numbers and measure unit, the use of apex and subscripts. Change “cells/ml” by “cells/mL” or “mol.L−1” by “M” (line 355).

Response: All of Section 5 "Material and Methods" has been revised and the numbers and units of measurement have been corrected. Please check the page 9-11.

Round 2

Reviewer 1 Report

Authors  have answered the questions and significantly improved the manuscript and presentation of the data. 

Minor remark to be considered. Authors obtained 3 samples of PLGA/PEI particles, but analysis of cytotoxicity, complexation with BJ and so on is presented only for particles with highest content of PEI. Either justification of the choise of these very particles as a system with the hieghest potential as immunoadjuvant should be presented or the data on the other systems should be added.

Author Response

RESPONSE TO THE COMMENTS OF THE REVIEWER #1

General comment:

- Authors have answered the questions and significantly improved the manuscript and presentation of the data.

“Minor remark to be considered. Authors obtained 3 samples of PLGA/PEI particles, but analysis of cytotoxicity, complexation with BJ and so on is presented only for particles with highest content of PEI. Either justification of the choise of these very particles as a system with the hieghest potential as immunoadjuvant should be presented or the data on the other systems should be added”.

Response: We appreciate the referee’s comment. In fact, the formulation choosing to studies was content 1:5 PEI:PLGA. Several studies show the high capacity of cationic nanoparticles or ionized systems to modulate the immune response. Our approach was using the polycation PEI as a functionalizing agent for inducing cationic charge on the surface of nanoparticles. Our study confirm the capability of our formulation as immunoadjuvants since the behavior of animals response were similar to conventional immunoadjuvant. We revised the main text manuscript to improve the explanation about our proposal according to the suggestions. Please find below the responses to the comments and an itemized list of changes in the manuscript highlighted in RED. Please check the lines 82-86; 177-179;216-221.

Reviewer 4 Report

The manuscript looks much improved now

Author Response

RESPONSE TO THE COMMENTS OF THE REVIEWER #4

General comment:

- The manuscript looks much improved now.

Response: We appreciate the referee’s comment. We believe that all the suggestions contributed for improving the clarity and quality of our message in the manuscript. We did a thorough job of re-evaluating the text to improve the understanding of our results.

Reviewer 5 Report

I'm not convinced by Figure 1C (SDS PAGE ANALYSIS). First of all decrease the size. Secondly, why if authors used comassie blu, the bands have different colours. Thirdly, please add some comment regarging the gel in the main text, including also some comment about empty lanes.

Author Response

RESPONSE TO THE COMMENTS OF THE REVIEWER #5

General comment:

- “I'm not convinced by Figure 1C (SDS PAGE ANALYSIS). First of all decrease the size. Secondly, why if authors used comassie blu, the bands have different colours. Thirdly, please add some comment regarging the gel in the main text, including also some comment about empty lanes”.

Response: We appreciate the referee’s comment. The different colors identified in the he last version of Figure was because we used silver nitrate (Morrissey, 1981) to improve the result and precisely verify the empty lines. As suggested, the method description was improved in the main text. In addition, the Figure resized was added and comments about gel and empty lanes addressed in the text. Please check the new Figure 1C and lines 189-194.

Please also see lines 302-303.

Morrissey, James H. “Silver Stain for Proteins in Polyacrylamide Gels: A Modified Procedure with Enhanced Uniform Sensitivity.” Analytical Biochemistry, vol. 117, no. 2, 1981, pp. 307–10, doi:10.1016/0003-2697(81)90783-1.